# Continuous monitoring and machine vision reveals that developing gerbils exhibit structured social behaviors prior to the emergence of autonomy

Catalin Mitelut[1,2�]*, Marielisa Diez Castro[2�again], Ralph E. Peterson[2], Maria Gonçalves[2], Jennifer Li[2], Madeline Gamer[2], Simon RO Nilsson[2], Talmo D. Pereira[3‡], Dan H. Sanes[2‡]

1 Netholabs, London, United Kingdom, 2 Center for Neural Science, Department of Biology and Department of Psychology, New York University, New York, New York, United States of America, 3 Salk Institute for Biological Studies, La Jolla, California, United States of America,

☾ These authors contributed equally to this work.
‡ TDP and DHS are co-last authors on this work.
* mitelutco@gmail.com

## Abstract

Investigating social and independent behavior structure in early life is critical for understanding development and brain maturation in social mammals. However, this investigation necessitates monitoring animals over weeks to months often with subsecond time resolution creating challenges for both lab studies focused on brief observation periods and field studies in which animal tracking can be imprecise. Here we used machine vision and two-week long continuous behavior recordings of families of gerbils, a highly social rodent, in large, undisturbed home environments to quantify the behavioral development of individual pups. We discovered that individual pups exhibited complex social behaviors from the first day they left the nest including a preference for interactions with siblings over parents. Critically, independent behaviors such as foraging for food and water emerged several days later, each with a stereotyped temporal trajectory. Analysis of individual animal development confirmed the quality of our tracking methods and the stability and distinctness of each behavioral measure. Our work supports a model in which early and sustained social interactions may be supportive of solitary exploration for physiological needs. This model suggests that understanding the development of behavioral independence as well as maturation of sensory and motor systems in social rodents such as gerbils may require integration of social behavioral knowledge earlier than typically considered.

## Introduction

At the center of mammalian behavioral development lies the emergence of agency [1]: the capacity to self-organize one's actions towards goals allowing individuals to

**Data availability statement:** All datasets including figure metadata have been deposited with Dryad and available at https://doi.org/10.5061/dryad.pc866t1vk. Software processing pipelines are publicly available and include SLEAP (https://github.com/talmolab/sleap), SimBA (https://github.com/sgoldenlab/simba) and custom postprocessing code available at https://github.com/catubc/gerbil.

**Funding:** This work was supported by the National Institutes of Health NIH R01DC020279 and R34 R34DA059513 to DHS. The funders had no role in study design, data collection and analysis, decision to publish, or preparation of the manuscript.

**Competing interests:** The authors have declared that no competing interests exist.

**Abbreviations:** BLA, basolateral amygdala; HPC-MEC, hippocampal and medial-entorhinal-cortex axis; NAc, nucleus accumbens; PAG, periaqueductal gray; PCA, principal component analysis; ROI, regions-of-interest.

survive and thrive [1,2]. These goals include the onset of foraging for resources, and learning through environmental affordances [3]. In contrast, family behavior in many species is initially dominated by intense social interactions between parents and offspring. Despite this, longitudinal studies in rodents are rare [4], especially ones that seek to understand the role played by social and family interactions during early development.

The development of agency increases with exploration outside the nest, and culminates with dispersal during which familial social bonds are severed [5–10]. These behavioral changes overlap substantially with the maturation of sensory (eye/ear development: [11]; sniffing: [12]), motor [13], and spatio-cognitive systems [14]. Social and environmental factors may impact the emergence of agency, including the attraction of pups to adults that are outside of the nest [15–17], a decline in nest signals that promote affiliation [18–20], and instrumental learning about social variables [21].

Studying the developmental relationship between social and solitary behaviors can be carried out in rodent species with strong family social relationships such as the Mongolian gerbil. Here, we operationalize social behaviors as those that involve approach or close physical proximity, and solitary behaviors as those performed autonomously, without apparent coordination or cooperation with other individuals. Gerbils live as multi-generational families with pair-bonded founder pairs, coordination between male and female parents during pup rearing, and alloparenting by older siblings [22–26]. Furthermore, they display family-specific vocal repertoires, suggesting the presence of a dialect that may represent kinship [27]. In the laboratory environment, gerbil pups largely remain in the nest until postnatal day (P) 16 and, while they increasingly leave the nest from P18–P22 [28], they remain attracted to maternal nest odors until at least P42 [29].

To gain new insights into how autonomous and social behaviors develop within family groups, we used uninterrupted home-cage video recordings in concert with a machine learning framework to continuously track individual Mongolian gerbil family members (2 adults and 4 pups) for two weeks. We began recording when animals first left the nest (approximately P15) and collected >1,000 hrs of video data (approximately 100 million video frames) across three cohorts of animals, and implemented and adapted pose estimation tools [30], behavior classifiers [31], and custom algorithms for behavior classification. Our results reveal that the presence of complex but stereotyped structured social behaviors early in development precede the development of solitary exploration for food and water. These results point to a more complex contribution of social behaviors to rodent development.

## Results

### Tracking behaviors of individual gerbil family members in home-cage environments

**Capturing individual gerbil behavior across weeks with subsecond temporal precision.** All analyses are based on 12 juveniles and 6 adults in three gerbil family cohorts from postnatal days (P) 15–30 (Fig 1a; see also "Methods"). Prior to data

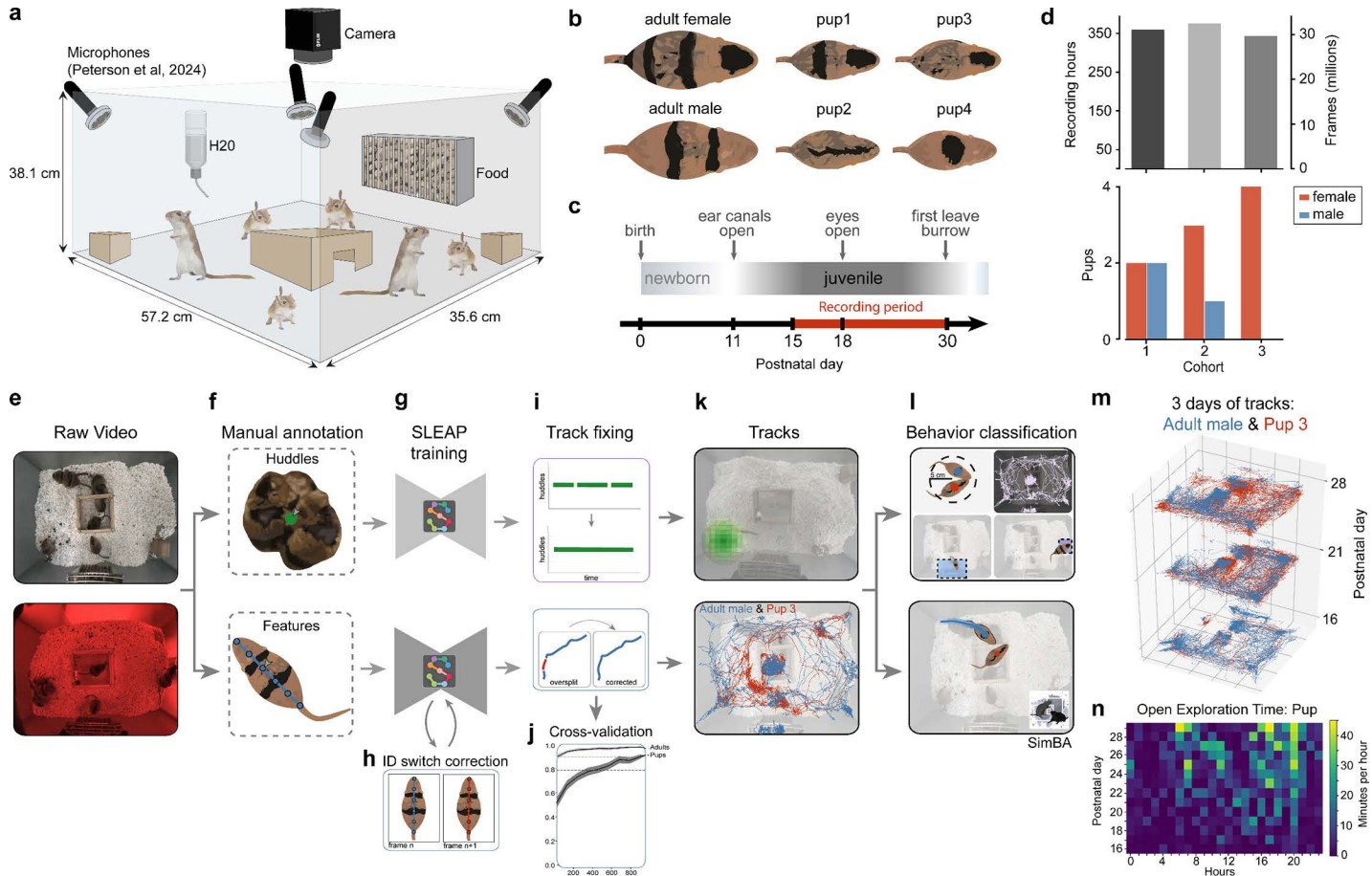

**Fig 1. Capturing individual gerbil behavior across weeks of development with subsecond precision. (a)** Homecage and recording setup using continuous, synchronized video (single overhead camera at 25 fps) and audio (4 ultrasonic microphones at 125 kHz) recordings in environments with *ad libitum* access to food and water and enrichment consisting of one wooden hut and two blocks. **(b)** Individual gerbil identification based on distinct shaving patterns. **(c)** Recording periods of 15 days from postnatal (P) day 15 to 30. **(d)** Top: number of recording hours and frames per cohort; Bottom: number of female and male pups per cohort (bottom). **(e–l)** Workflow of animal tracking and behavior analysis. **(e)** Acquisition of raw video in a 12 hrs light/dark (infrared) cycle. **(f)** Manual annotation of two categories: huddles (2 or more gerbils huddling together in the nest area) and features (individual animals with a 6-node skeleton). **(g)** Each light condition (light and dark) and cohort required separate training resulting in six feature models and six huddle models. **(h)** Method for identification of the identity switches between consecutive frames. **(i)** Method for interpolating small missing tracks based on id-swapping cues. **(j)** Cross-validation of model performance (note: chance is 1/6 = 0.17). **(k)** Visual representation of nest location and animal tracks. **(l)** Top: sketch of behaviors classified using heuristics (e.g., time spent in a region of interest and distance traveled); Bottom: sketch of approach behaviors captured using SimBA classifier. **(m)** Two animal tracks (adult male and pup) across 3 days of development. **(n)** Example ethogram of open exploration time per hour for one pup across 15 days of development. The metadata can be found on Dryad: DOI): https://doi.org/10.5061/dryad.pc866t1vk.

collection, all gerbils were shaved with a distinct pattern to facilitate identification of individuals by animal tracking machine vision algorithms (Fig 1b). Families were placed in an isolated enriched home environment at P14, and we began video recordings on P15, the age at which gerbil pups begin to leave their nest [32]. Data collection ended on P30, the age at which juveniles are weaned in our colony (Fig 1c). Therefore, the behavioral analysis of each juvenile and adult is based on more than 30 million frames of video data per animal (Fig 1d).

**Tracking individual gerbil behaviors.** We used SLEAP [30] to identify individual gerbils and gerbil huddles across day and night cycles (Fig 1e; see also "Methods"). For tracking huddles, we trained SLEAP on manually annotated frames

with groups of 2 or more gerbils that were huddling at a stable location, and for tracking gerbils in the open field we annotated 6 body features along the spine of the gerbils on randomly chosen 1,000 frames from dozens of videos (Fig 1f). Because of idiosyncratic differences in shaving patterns and light conditions across the cohorts, we trained separate models for each cohort on day and night conditions, resulting in 6 body-feature models and 6 huddle models (Fig 1g).

We also developed methods for improving animal identity tracking: (i) an identification-switch detection pipeline that automatically located identity tracking errors on sequential video frames (Fig 1h; see "Methods"), (ii) a track fixing pipeline that interpolated short missing periods of huddling or individual tracks (Fig 1i, top), and (iii) an identity swap pipeline that detected and corrected obvious track identity swaps (Fig 1i, bottom). We used 10-fold cross validation to evaluate SLEAP *identity* accuracy and found a true positive rate of 0.97 (0.04 std) for adult animals and 0.91 for pups (0.05) (Fig 1j; see also S1 Fig). Note that chance identification is 0.17. Errors in *feature* (i.e., swapping of body parts within an animal) were extremely low (<0.1%), but we used centroids to represent animal locations that were less sensitive to such errors (see "Methods"). Using our pipelines, we were able to track the huddling (Fig 1k, top) and open field behaviors of multiple animals across weeks of continuous recording (Fig 1k, bottom).

**Classifying animal behavior.** We implemented two types of methods for classifying behavior: heuristic-based methods and supervised classification methods (Fig 1l). For heuristic-based classifications we used Regions-of-Interest (ROIs) to identify when animals were near the food hopper or waterspout regions, and proximity metrics for detecting when gerbils were within 5 cm of one another (Fig 1l, top; see "Methods"). For approach behaviors we used a customized version of SimBA [31] to train a classifier to capture approach behaviors from manually labeled examples (Fig 1l, bottom) (see "Methods"). Using our pipelines, we were able to track individual gerbils over periods of two weeks with subsecond predictions (Fig 1m) and generate ethograms that captured dynamics from postnatal day over several behaviors (Fig 1n for an example).

In sum, we found that unique shaving patterns, and tracking methods with customized algorithms enabled the tracking of multiple individuals over weeks of behavior. These tracking results enabled us to characterize the development of individual and group behaviors in gerbil pups during the period of development when animals adopt independent repertoires, as described below.

## The emergence of autonomous behaviors

Using the processing pipelines and heuristics described above, we focused on tracking the development of autonomy, i.e., the development of capacities to regulate and sustain oneself *independently* of other animals' behaviors, including the parents.

**Diverse and dynamic development of territory exploration in gerbil pups.** We tracked the amount of time gerbils spent exploring their habitat outside the nest, in the food hopper region, and in the waterspout region, as well as the total distance traveled per day (Fig 2a; see also "Methods"). As we were interested in the maturation of behavior, we evaluated pup behavior relative to the average adult behavior recorded when pups were P29 (Fig 2b). We selected P29 as the adult reference date as we found behaviors of adults stabilized by this period. Thus, all graphs represent the adult mean as a value of 1.0, and pup values are expressed as a proportion relative to the adult mean. We differentiated between periods during which behaviors were (i) *stable* (Fig 2b – colored lines) or (ii) *rapidly developing* (Fig 2b – triangles). For each behavior, we defined a postnatal day as a period of *rapid development* when that day was statistically different from the previous 2 days and the following day (2-sample K-S test; $p$-value < 0.05; see also "Methods"). We also calculated the spatial overlap of each of the behaviors by using principal component analysis (PCA) first to reduce the dimensionality of the multi-day behavior trajectories and then calculated the convex hull overlaps across behaviors (Fig 2c; see "Methods"). We used 3 PCs to compute the 3d intersection of the behaviors as 3PC provided approximately 80% (or more) variance explained for all the behaviors (see "Methods"). We found this method was more easily interpretable than Kullback–Leibler divergence or Mahalanobis distances, which do not directly quantify overlap as a percentage.

 

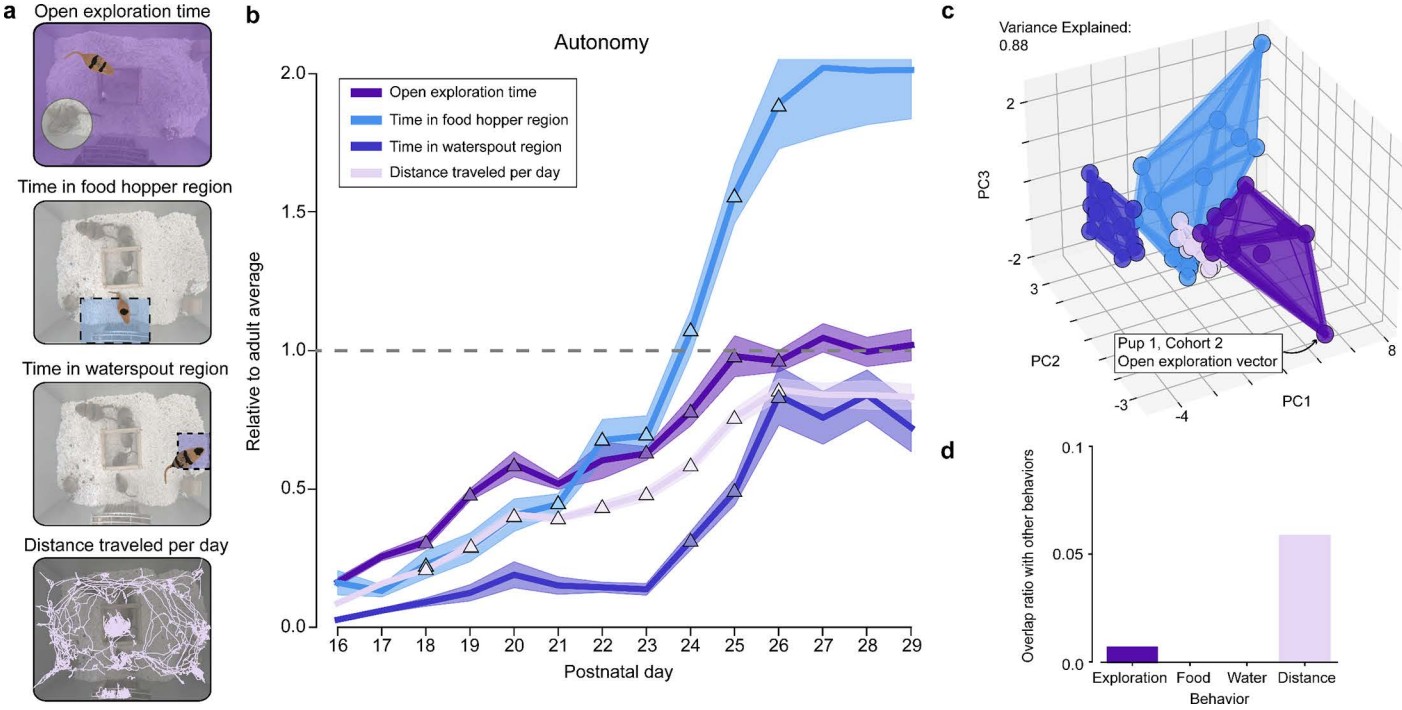

**Fig 2. The development of independent exploratory behaviors. (a)** Sketches illustrating autonomous behaviors including open exploration time, exploration of food hopper and waterspout regions, and total distance traveled. **(b)** Average environment exploration trajectories in gerbil pups relative to the adult average on P29 (lines) and standard error (SEM; shading) with indication of rapid development days (triangles; see "Methods"). **(c)** Convex hull overlap analysis of individual animal multi-day trajectories with intersections calculated in 3D PCA space (i.e. PCs 1–3) with an example of a single pup trajectory for open exploration time. **(d)** Overlaps in PCA space from **(c)** show limited inter-behavior overlap. The metadata can be found on Dryad: DOI): https://doi.org/10.5061/dryad.pc866t1vk.

We found that pup *distance traveled per day* started at 0.09±0.02 of the adult mean (or 8.01±1.53 m/day) and increased continuously until P26, where it stabilized near adult levels at 0.85±0.13 of the adult mean (or 78.98±9.91 m/day). Pup *open exploration time* followed a similar trajectory, starting at 0.16±0.05 of the adult mean (or 0.88±0.32 hrs/day) and reaching 0.96±0.13 of the adult mean (or 5.17±0.84 hrs/day) by P25 (Fig 2b). We note that both distance traveled and environment exploration time underwent rapid development for most of the days between P17 and P25. These findings on open environment exploration suggest a steady and continuous maturation rate with pup behavior becoming similar to adults by P26, 11 days after gerbil pups leave the nest.

In contrast, food hopper exploration increased slowly from P16 at 0.16±0.15 of the adult mean (or 0.14±0.10 hrs/day) to 0.45±0.25 of the adult mean at P21 (or 0.44±0.13 hrs/day). There was a second period of rapid development between P22 and P26, and stabilizing from P27 to P29 at 2.02±0.68 of the adult mean (or 1.92±0.25 hrs/day), more than double the adult average (Fig 2b). This latter increase in time at the food hopper indicates that juveniles must devote a fractionally greater amount of time consuming calories as they transition to independence.

Waterspout region exploration also displayed three stages, but with a much later trajectory than food: (i) an initial stage in which pups spent less than 0.10 of the adult average from P16 at 0.03±0.01 of the adult mean (or 0.007±0.004 hrs/day) to P23 at 0.14±0.07 of the adult mean (or 0.04±0.02 hrs/day), (ii) a second period of rapidly increasing waterspout exploration from P24 at 0.31±0.11 of the adult mean (or 0.07±0.02 hrs/day) to P26 at 0.84±0.35 of the adult mean (or 0.19±0.07 hrs), and (iii) a third period of stability from P27 to P29 where gerbil pups spent 0.77±0.30 of the adult mean

(or 0.18±0.02 hrs/day). The rapid increase from P24 to P26 may reflect a discovery of water sources and/or a rapid reduction in the time spent nursing.

**Distinct inter-behavior developmental trajectories of territory exploration in gerbil pups.** We next investigated whether individual pup behaviors were distinct from one another. We again calculated the spatial overlap as described above (Fig 2c; see "Methods"). Across the four behaviors tracked we found only minimal overlap between open exploration time (0.73%) and distance traveled per day behaviors (5.9%) (Fig 2d). Correct animal identity was important to this finding as shuffling identity for trajectory (S2 Fig) and PCA space visualizations (S3 Fig) decreased the separation and uniqueness of trajectories. Additionally, correlation analysis also revealed the importance of correct animal identity (S4 Fig). We also note that some within-cohort clustering of behavior is present, but this behavior clustering does not appear to affect our main findings (S6 Fig).

In sum, we found that across four behavioral measures, we could identify distinct gerbil pup developmental time courses that were different from each other, and from those displayed by the parents. We note that there was some within-cohort similarity between trajectories and further analysis could be carried out to investigate the effects of this. Food and water consumption developed on different timelines and had their own unique periods of rapid change, suggesting complex interactions between drives for nutritional needs and independent exploration of the environment.

## Structured social behaviors precede the development of autonomous behaviors

We next sought to characterize pair-wise and group social behaviors to capture dependencies between individual family members (Fig 3). We defined social behaviors as those that *involved another animal* such as time spent huddling in the nest or approaches between two animals. As we were interested in the maturation of behavior, we again evaluated behaviors relative to the average adult behavior recorded when pups were P29 (as in Fig 2).

**The development of nest-related social behavior in gerbil pups.** With respect to *nest and huddling behaviors*, we calculated two metrics: (i) time that pups spent huddling in the nest; and (ii) amount of times adults followed pups out of the nest to capture the tendency of adults to retrieve or care for younger pups (Fig 3a; see also "Methods"). We found that pups spent more time than adults in the nest on P16 at 1.21±0.02, 0.02 of the adult mean (or 23.12±0.02 hrs/day) and this value reached and stabilized at the adult average by P24 at 1.04±0.06 of the adult mean (or 19.80±1.05 hrs/day) (Fig 3b). Similarly, pups were followed out of the nest by parents for more time than parents followed each other on P16 at 1.51±0.02 of the adult mean. This nest following behavior also stabilized around P24 at 1.07±0.12 (Fig 3b) (Note: the two behaviors had similar developmental trajectories with nest exits having higher variance and only 0.06 overlap, and nest huddling having very low variance and 0.46 overlap).

With respect to *pairwise social exploration*, we tracked the amount of time pups spent together with each other or with other adults (Fig 3c). We found that pups tended to spend more time near their siblings than with the adult parents (Fig 3d). In particular, at P16 pup–pup pairwise exploration was 0.29±0.04 of the adult mean (or 9.15±4.15 min/day), but increased rapidly to adult levels within a few days reaching 1.00±0.12 of the adult mean (or 28.03±7.55 min/day) by P19 and stabilizing on P29 at 1.23±0.05 of the adult mean (or 33.67±1.51 min/day). In contrast, adult–pup pairwise exploration time started on P16 at 0.24±0.02 of the adult mean (or 5.35±1.88 min/day) and only reached a peak around P26 at 0.68±0.05 of the adult mean (or 16.02±4.03 min/day). These findings suggest that pups interacted more with other pups throughout development as compared to adult interactions (e.g., pup–pup interactions were nearly double those of pup–adult interactions across the full observation period). Taken together with pairwise approaches, the pairwise socialization times results suggest that, while pups approach other pups fewer times per day during early development (Fig 3d), they spent more time together following an approach (Fig 2f).

For *approach* behaviors, we found a similar pattern between pup–pup and pup–adult approaches, with some important differences (Fig 3e). Pup–pup approaches were approximately 2-fold lower on P16 at 0.10±0.08 of the adult mean (or 3.52±2.97 approaches/day), as compared to pup–adult approaches at 0.24±0.15 of the adult mean (or 10.08±6.38

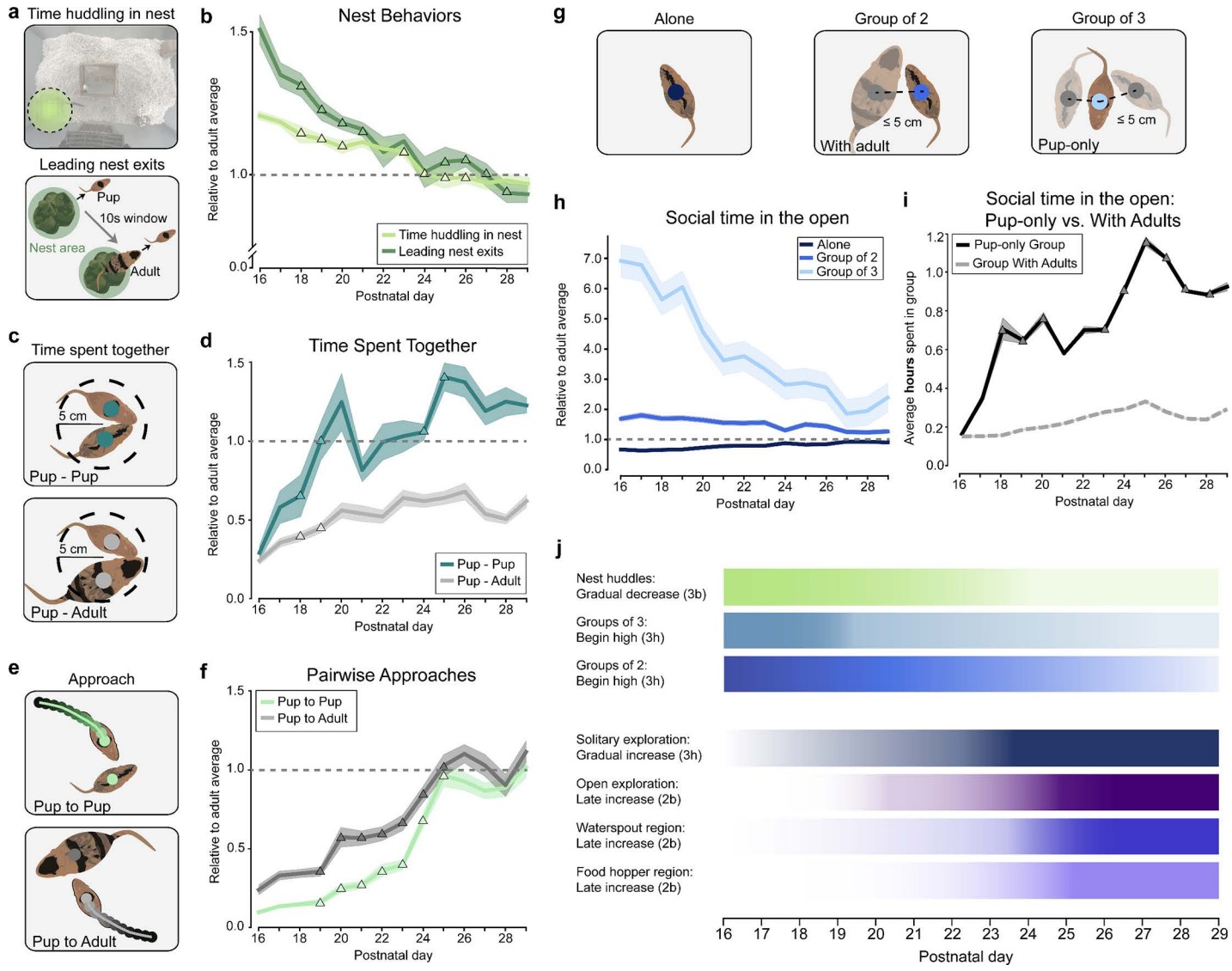

**Fig 3. The development of social behaviors. (a)** Sketches illustrating nest behaviors (time huddling in nest) and sequences of adults following pups out of the nest (within 10 s). **(b)** Average huddling time in the nest (light green) and leading nest exits (dark green) relative to the adult average on P29 (shading-SEM; triangles rapid development days – see "Methods"). **(c)** Sketch illustrating pairwise proximity behaviors. **(d)** The average time of pup time spent in close proximity with each other (turquoise) and with the adults (gray) relative to the adult average on P29 (shading-SEM). **(e)** Sketches of pairwise approaches. **(f)** The average number pup approaches towards other pups (green) and adults (dark gray) relative to the adult average on P29 (shading-SEM). **(g)** Sketches of single or multi group proximity. **(h)** Average time spent in the open by pups as individuals (dark blue) or groups of two (blue) or groups of three (light blue) relative to the adult average on P29 (shading-SEM). **(i)** Average hours spent in groups with pups (black) vs. groups with adults (dashed gray line). **(j)** Summary of rapid development periods (colors) across all behaviors tracked. The metadata can be found on Dryad: DOI): https://doi.org/10.5061/dryad.pc866t1vk.

approaches/day) (Fig 3f). Both types of approaches increased gradually, with pup–pup approaches undergoing rapid development from P20 at 0.25 ± 0.13 of the adult mean (or 11.14 ± 5.44 approaches/day) to P25 at 0.96 ± 0.23 of the adult mean (or 39.76 ± 16.21 approaches/day) before leveling off to near adult levels: P26 to P29 averages 0.92 ± 0.18 of the adult mean (or 38.29 ± 13.29 approaches/day). Pup–adult approaches also underwent rapid development from P20 at 0.57 ± 0.22 of the adult mean (or 28.21 ± 8.95 approaches/day) to P25 at 1.03 ± 0.21 of the adult mean (or 35.67 ± 9.40

approaches/day) and leveling off to adult levels (P26–P29: 1.04 ± 0.32 of the adult mean, or 44.18 ± 12.73 approaches/day for pup–adults, compared to 43.33 ± 9.50 for adult–adults).

Lastly, we calculated *group socialization* time as the duration spent in groups of two or three gerbils (Fig 3g–3i). We found that pups socialized in groups of three gerbils 6–7 times more frequently than adults did in early development when they left the nest (i.e., as early as P16) and leveled off at around 2 times as much by P29 (Fig 3h). Looking specifically at pup versus adult groups, we found that as early as P18 pups preferred pup groups over groups with (any) adults by a factor of 3–5 fold (Fig 3i). This pattern suggests an early drive for socialization between siblings, rather than partner preference being monolithic across pup and adult family members. Over behaviors, as expected, we find high correlations and anti-correlations due to the monotonously increasing or decreasing angle of the behavioral trajectories (see S5 Fig).

**Structured social behaviors precede the development of autonomy.** As illustrated in Fig 3j, social behaviors in the open field (e.g., proximity of 2–3 pups) were prominent as pups first left the nest at P16–P18. In contrast, biophysical agency (e.g., food and water acquisition) displayed a later onset, only increasing substantially at P22–P24. The delay of nutrition-seeking behaviors is perhaps explained by Mongolian gerbils having a longer period of parental care, including biparental care [33,34]. We also note that rapid development after P26 stopped for all but large group socialization suggesting that pup development across the several axes we studied was largely complete, at least given the limitations of a closed environment.

In sum, we found that both social behaviors and environmental exploration had unique developmental trajectories (Fig 3j). We also observed distinct periods of maturation during which pup behavior changed rapidly over a few days. Since inter-behavior differences were substantially greater than inter-animal differences, we suggest that different classes of behavior emerge on distinct timescales that correlate with the maturation of several components of the nervous system in rodents between P15 and P30 (see "Discussion").

### The development of multi-animal social state dynamics

We next evaluated the development of multi-animal social behaviors by focusing on the complexity of social states and configurations in the home cage. We were particularly interested in quantifying the complexity of multi-animal social configurations in their home environment. We partitioned the environment floor into a 5 by 4 grid and defined a social state at each recording time point as the number of animals at each location (Fig 4a, 4b) (see also "Methods"). This procedure enabled us to generate a network-based analysis where each time point in our recording generated a 20 Dimensional (20D) social state vector and we could generate a network connectivity matrix by connecting all social states that occurred sequentially (Fig 4c; see also "Methods"). The network connectivity matrix enabled us to generate graphs that represented social state dynamics in our cohorts (Fig 4d).

**The number of social-state configurations increase across development for groups of animals.** We calculated a social-state graph for single animals, pairs (*n* = 2), all pups (*n* = 4) and all animals (*n* = 6) for each day of our study (Fig 4e, 4f). We first evaluated graph sizes that represent the number of unique social configurations in each graph (Fig 5e). We found that between P16 and P19, the graph sizes that grew most were all-pup graphs: P29/P16 ratio = 5.37, and all-animal graphs: P19/P16 ratio = 4.24 (Fig 4f). In contrast, social state complexity for pup–pup graphs increased much less: P29/P16 ratio = 1.61, and adult–adult complexity was mostly stable over time: P29/P16 ratio = 1.09. These differences suggest that social behavior complexity increased the most for larger groups of animals in ways possibly not accounted for solely by pair-wise social complexity increases alone. Additionally, consistent with our findings for social states in animal-pairs, we found that social state networks for larger animal groups (i.e., 4 and 6) leveled off at around P24 (Fig 4f).

**The number of social-state paths increases across development for groups of animals but not adult pairs.** We next calculated the number of cycles for each day and animal grouping (Fig 4g, 4h). The number of cycles can be viewed as the number of unique paths in the social state space that animals can traverse over time (Fig 4g; note we did not count repeated cycles). We found that both all-pup and all-animal groupings showed a substantial increase in the number

PLOS Biology

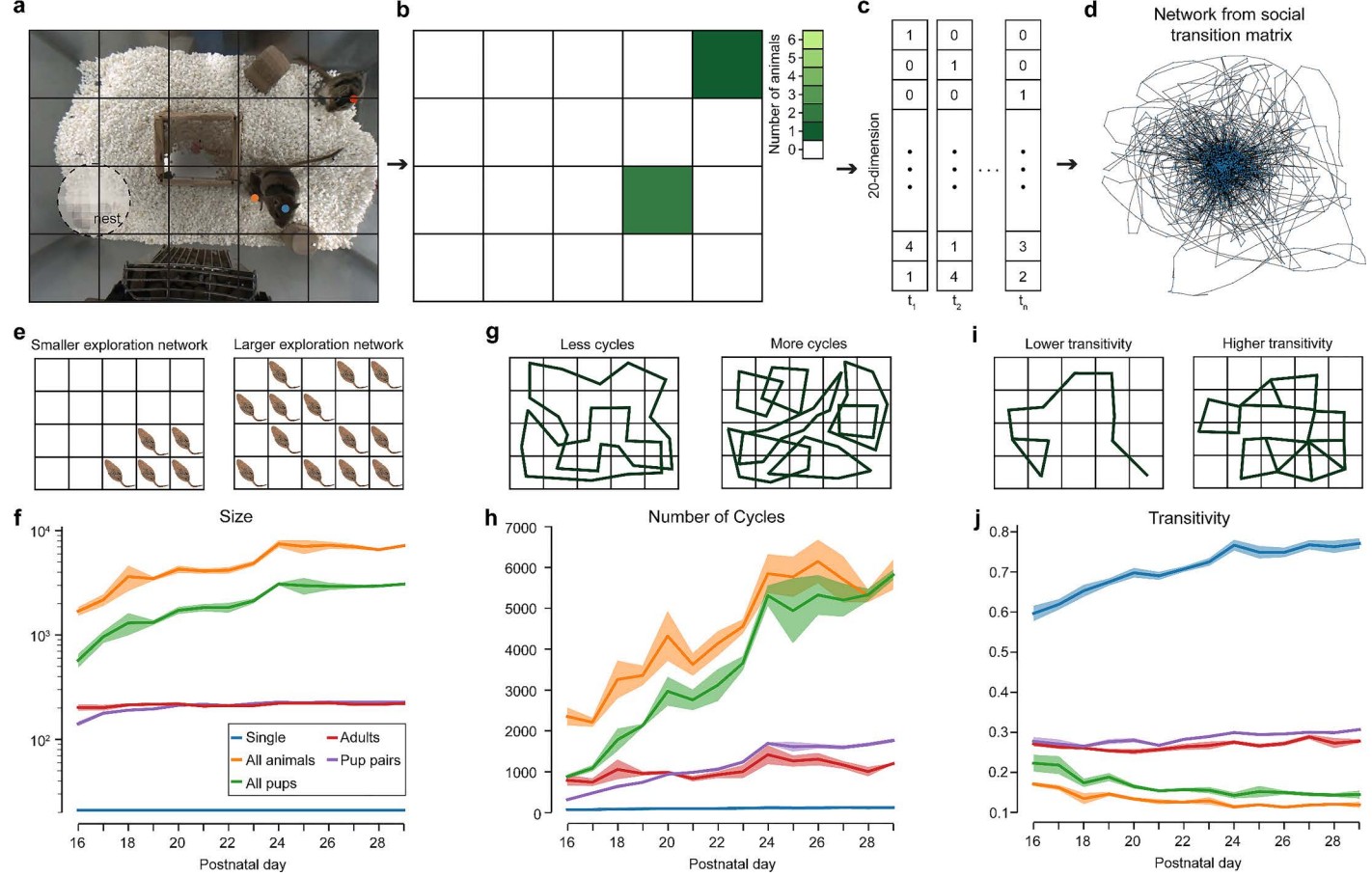

**Fig 4. The development of social behavior complexity. (a)** Partitioning the home cage environment into a 5 × 4 social-state grid. **(b)** Binarization of social states in **(a)** by the number of gerbils in each grid square. **(c)** Converting social-states to sequences of 20D vectors (note: vectors do not correspond to panel **(b)**). **(d)** Network graph obtained from social-state transitions for 1 day. **(e)** Sketch depicting behavior of a single gerbil that would generate smaller vs. larger sized networks (note: multi-animal configurations have more than 2 dimensions and are not simple to visualize; see "Methods"). **(f)** Network sizes vs. development for single and different groups of gerbils (note logarithmic scaling of *y*-axis). **(g)** Sketch depicting behavioral paths of a single gerbil that would generate graphs with less vs. more cycles. **(h)** Number of graph cycles vs. development. **(i)** Sketch depicting path of a single gerbil that would generate graphs with lower vs. higher transitivity. **(j)** Transitivity metrics vs. development. The metadata can be found on Dryad: DOI): https://doi.org/10.5061/dryad.pc866t1vk.

of cycles over development up to P24 when we observed qualitative leveling off as for network sizes (all-pup: P29/P16 ratio = 6.54; all-animal: P29/P16 ratio = 2.47). Similarly, pup–pup pairwise social states had a significant increase over development: P29/P16 ratio = 5.44. In contrast, the adult–adult social states had very similar numbers of cycles across development: P29/P16 ratio = 1.51. Interestingly, average path length for all groups appears to be relatively stable over development, or slightly decreasing, suggesting that individual social-state dynamical paths were not getting significantly larger – but were rather increasing in number, likely driven by new areas and configurations to explore. Taken together, the developmental dynamics of the number of cycles suggest that novel social-state configurations are generated during the period of P16 to P24 and are driven by sibling behavioral development with a smaller contribution from adults.

**Social exploration increases across development for large groups, but not pairs or single animals.** Lastly, we calculated the transitivity of each social configuration network (Fig 4i, 4j). Transitivity refers to the overall likelihood that a node has interconnected adjacent edges (i.e., the ratio of possible triangles in the graph) (Fig 4i). In our paradigm,

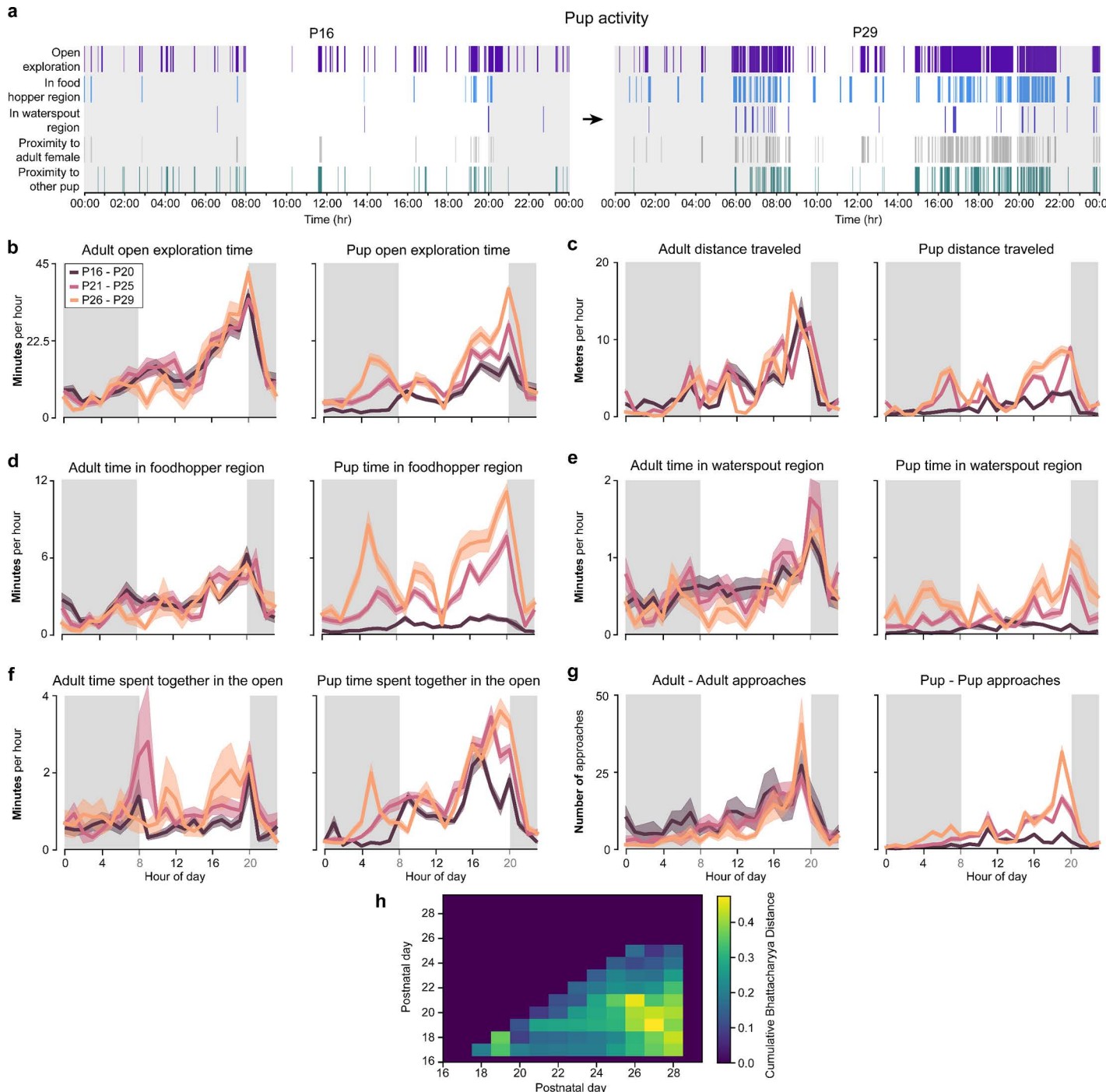

**Fig 5. Circadian rhythms in adult and developing gerbils. (a)** Behavior ethograms for one cohort showing substantial increases in the volume of behaviors between P16 and P29. **(b)** Average adult (left) and pup (right) open exploration time per hour (lines) and SEM (shading), across three developmental periods: P16 to P20 (dark purple), P21 to P25 (pink) and P26 to P29 (light orange). **(c)** Average adult (left) and pup (right) distance traveled per hour and SEM, across the three developmental periods. **(d)** Average adult (left) and pup (right) time spent in the food hopper region per hour and SEM, across the three developmental periods. **(e)** Average adult (left) and pup (right) time spent in the waterspout region per hour and SEM, across the three developmental periods. **(f)** Average time in proximity for adults with adults (left) and pups with pups (right) per hour and SEM, across the three developmental periods. **(g)** Average adult to adult (left) and pup to pup (right) approaches per hour and SEM, across the three developmental periods. **(h)** Cumulative Bhattacharyya distance (colors) for computing the largest circadian separation based on pup time in the food hopper region (see also "Results"). The metadata can be found on Dryad: DOI): https://doi.org/10.5061/dryad.pc866t1vk.

transitivity can be viewed as how comprehensively each social configuration is explored by the animals. High transitivity values would indicate that gerbils spend more time in individual configurations with slight alteration – whereas low transitivity values indicate that gerbils transition through more diverse social configurations without spending time exploring individual configurations. In simpler terms, high transitivity could be viewed as a *tendency to exploit* locally specific social configurations whereas low transitivity could be viewed as a *tendency towards exploring* more novel and different social configurations.

We found that single animals showed an increase in transitivity over time: P29/P16 ratio = 1.29; suggesting a tendency for single animals to exploit existing paths rather than pursuing new ones. In contrast, pairs of animals (adults and pups) had slight tendencies towards exploration: pup pairs P29/P16 = 1.10; and adult pairs P29/P16 = 1.03. In contrast, all-pup and all-animal groups had significant trends towards increased exploration of novel social configurations: all pups P29/P16 = 0.65; and all animals P29/P16 = 0.70. Taken together, these results suggest that when in larger groups animals tended to seek increasingly novel types of social-state configurations, essentially exploring the social state space more. Whereas as individuals or pairs, they did so much less. These findings suggest that increased group sizes could be supportive of environmental exploration and that single animals or smaller groups tend to use more stereotyped or repeated trajectory patterns. However, additional computational analysis including more in-depth shuffling and controls would provide a more robust test of this result.

## The development of circadian rhythm based behaviors

We next evaluated the correlation between the animal behaviors and the day:night cycles. In particular, we sought to quantify significant changes at the transition between light cycles (Fig 5a). For these analyses, all values are presented separately for both pups and adults as absolute values.

**Diurnal and nocturnal behaviors.** Across most behaviors tracked we found that both adult and developing pups had similar day:night cycle biases in behavior times (Fig 5). For example, the proportion foraging time near the food hopper for adults between day and night was 0.57 ± 0.01 (day cycle) versus 0.43 ± 0.01 (dark cycle) while for pups it was 0.60 ± 0.03 (day cycle) and 0.40 ± 0.03 (night cycle) (Fig 5d). These differences suggest a slight bias towards food consumption during the day. In contrast, the proportion of total time spent near waterspout for adults was evenly split between night and day, i.e., 0.50 ± 0.01 for day cycle and 0.50 ± 0.01 for night cycle, and similarly for pups: 0.49 ± 0.02 for day cycle and 0.51 ± 0.04 for night cycle (Fig 5e).

**Genetically versus environmental driven diurnal behavior structure.** Our observations of family development (Figs 2 and 3) suggested three stages: an initial period up to P19, a longer period up to P26 and a final stage from P26 to P29 during which all pup behaviors stabilized. Accordingly, we used this qualitative observation to initialize our analysis of development of circadian rhythms, but also sought to optimize these transition periods (see Fig 5h; "Methods").

In adults, we observed both nocturnal and diurnal behaviors, with a slight bias towards diurnal behavior and a systematic increase in the amount of behavior during daytime that peaked between 19:00 and 20:00, just prior to the night cycle onset at 20:00. This pattern was present for most developmental periods in adults and for most behaviors studied including exploration time (Fig 5b), distance traveled (Fig 5c), food hopper proximity (Fig 5d), waterspout proximity (Fig 5e) and adult–adult approaches (Fig 5g, left panel). It was absent, or very weak in adult–adult proximity time (Fig 5f, left panel).

For pups, we also observed a diurnal activity pattern over the day cycle across most behaviors. However, this pattern was present in all developmental periods only for exploration time (Fig 5b, right panel) and pup–pup time spent together (Fig 5f, right panel). For other behaviors, this pattern effect developed gradually in several behaviors: food hopper region exploration (Fig 5d), waterspout region proximity (Fig 5e), distance traveled (Fig 5c), and pup–pup approaches (Fig 5g, right panel).

**The development of circadian patterns.** We observed *circadian* behavior in both pups and adults for some, but not all, behaviors. In adults, *circadian* behavior was present in distance traveled (Fig 5c), weaker in waterspout region

proximity (Fig 5e), and mostly absent or very weak in food hopper region proximity (Fig 5d), social foraging time (Fig 5f), and adult–adult approaches (Fig 5g, left panel). (We note that our observation of a weak *circadian pattern* in waterspout region proximity is similar to Pietrewicz and colleagues) [35]. In pups, there were strong *circadian* patterns in exploration time (Fig 5b), distance traveled (Fig 5c), food hopper proximity (Fig 5d), and pup–pup proximity time (Fig 5f, right panel). Waterspout region proximity had only a weak *circadian pattern* (Fig 5e) and there were few and small pre-dawn peaks in pup–pup approaches (Fig 5g, right panel). Importantly, *circadian patterns* developed over time for food hopper, pup–pup time spent together, exploration time and distance traveled, being largely absent in early development (i.e., P16–P20). Interestingly, we note that qualitatively, pups appear to develop much more pronounced pre night-to-day cycle peaks generally around 4:00 and 5:00 than adults (night-to-day cycle change was at 8:00).

**Detecting the development of circadian rhythms.** We used a computational approach to agnostically identify the age at which behaviors transitioned from non-circadian to circadian patterns. In particular, we sought to determine the times of transition between: an initial period D1, a middle period D2 and a final period D3 (initialized as P16–P20, P21–P25, and P26–P29 above). We thus sought to determine the optimal two transition points, $t1$ and $t2$, determining the transition from D1 to D2 and D2 to D3, respectively, by maximizing the difference between the individual animal circadian trajectories in each of these periods (see "Methods"). We used food hopper region proximity as it provided the most obvious differences in development, and we found $t1 = 19$ and $t2 = 27$, yielding three developmental periods of D1: P16–P19, D2: P20–P27, and D3: P28–P29. (Fig 5h). We note that as this method is based on initialization heuristics (e.g., it assumes three developmental periods) different priors or assumptions may provide slightly different developmental windows. We note that this analysis relies on heuristics and assumptions noted above and additional confounds may explain these transitions rather than an innate circadian rhythm development program.

## Discussion

Using continuous recordings in an enlarged home-cage environment, we sought to quantify the development of social and autonomous behaviors in gerbil pups – described together as the development of agency – across a highly dynamic developmental window. To do so, we employed machine vision tools and developed methods for quantifying a set of unique developmental trajectories. Taken together, our findings show that, during the period from P15 to P30, gerbil pups transition from living almost exclusively in the nest to becoming highly independent.

**Pup behaviors develop along distinct time courses.** Across both autonomous and social behaviors, we observed stereotyped trajectories in development. Furthermore, the time courses tracking the development of single behaviors for pups from different cohorts were more similar to each other, than when compared with the time courses from the same pup but for different behaviors (Figs 2c and 3b, 3d, 3f, 3h, 3i). This pattern suggests that these classes of behaviors emerge on distinct timelines rather than being driven by random and/or individual processes. For example, foraging-based metrics such as time spent exploring and distance traveled per day each displayed a gradual increase between P16 and P26, at which time pups reached adult-like levels. In contrast, food proximity behavior displayed two phases, increasing gradually at first, but escalating dramatically from P22–P26, and reached 200% of the adult average. Similarly, water proximity behavior emerged slowly at first, but increased substantially on P24 and reached approximately 80% the adult average by P26. Taken together, these results suggest a slow and prolonged development of exploration behaviors with a rapid value-driven emergence of food and water exploratory behaviors around P22–P24. In fact, the rapid increase in pup food and water exploration coincides with a decline in adult female time spent at the food hopper (26% decrease after P24) and waterspout (41% decrease after P24).

Pup open exploration time increased from <1 hr at P16, to >5 hrs by P25. However, pup social behaviors predominated as they first began to leave the nest. As compared to adults, pups preferentially socialized in groups of 2 or 3, and preferentially associated with one another, rather than a parent, from P16–17 onwards (Fig 3d, 3h, 3i). For example, pups spent up to 7 times more time in groups of 3 than did their parents in the first few days after leaving the nest.

This observation leads us to speculate that the early bias towards social interactions may prepare pups for later solitary activity.

**Early social interactions support the transition to independence.** Juvenile and adolescent animals often display a rich array of social interactions with their peers, and this may serve as the substrate for the emergence of adult skills [36–38]. With few exceptions (e.g., [39,40]), the long-term effects of experimentally manipulating early social experience have been carried out in rodents. For example, depriving pre-adolescent rats of social play for two weeks leads to decreased social activity in adulthood [41, 42]. Furthermore, juvenile play during a developmental critical period can induce a preference for the environment in which the social interaction occurred [42–45], and this effect is diminished when pups are treated with a drug that interferes with social play [46]. Our descriptive data is broadly consistent with a constructive role of early social experience. When gerbil pups begin to leave the nest, they preferentially retain sibling interactions (Fig 3d, 3h, and 3i), which then leads to a dramatic increase in open exploration time (Fig 2b).

**Behavior development is plausibly supported by the maturation of spatial-cognition and motor systems.** A systematic finding in our study is that all social and foraging behaviors commence at near zero levels relative to adults at P16, and mature with distinct time courses between P16 to P26. This period coincides with the maturation of neural pathways that subserve spatial memory, especially in the hippocampal and medial-entorhinal-cortex axis (HPC-MEC). In mice, there are several systems that develop during this period: formation and maturation of head direction cells (P10–P20), place cells (P14–P40), boundary responsive cells (P16–17) and grid cells (P20–P21) (Tan and colleagues [47]). In rats, hippocampus dependent learning has also been shown to commence around P18 for water maze learning (e.g., visual platform – P17–20 – and hidden platform – P19–21 – learning), and T-maze learning (P19–P42) [14]. In fact, social encoding of family membership may, itself, reside in the hippocampus [48]. The behaviors that we observed are not closely correlated with the onset of sensory transduction. The onset of transduction displays a similar sequence in most vertebrates, progressing from somatosensory to vestibular, olfactory, auditory and, finally, to the visual system [49]. However, a prolonged development of central sensory processing has been demonstrated for many mammals [50]. For example, as compared to adults, gerbil auditory perceptual skills and auditory cortex processing remain immature during early adolescence [51,52]. Therefore, one plausible interpretation is that the highly dynamic and specific periods of development that progress from a minimal state to near-adult levels between P16 to P29 are driven, in part, by maturation of HPC-MEC axis and central processing of sensory information.

A second plausible interpretation is that the changes and stabilization we observe arise as the motor cortex matures. In particular, the motor cortex is still relatively immature (e.g., as compared to sensory cortices), with an estimate that M1 starts producing a significant outflow only after postnatal day 20 [13]. Cortical motor outflow appears to begin at approximately P24 in rats [53–55], and there is a significant increase in stimulation-related motor activity after this age. For example, M1 activity recorded in vivo in P19–23 rats has yet to exhibit activity before wake or sleep movements [56]. Therefore, an increase in exploration may also be tied to the maturation of the motor cortical pathway.

While the maturation of the HPC-MEC axis, sensory, and motor cortices may contribute to behavioral changes during P16 to P29, it will be essential to consider other regions involved in development and threat-related behaviors. The BLA (basolateral amygdala), septum, and PAG (periaqueductal gray) play crucial roles in exploration and avoidance of threats during this time [57]. Additionally, regions involved in motivation, such as the nucleus accumbens (NAc), likely influence behavior, especially in relation to hunger, growth spurts, and pubertal changes, which may affect exploratory behaviors [58]. It is also worth noting that non-rod or cone-mediated phototaxis in young rodents [59], may wane during development, potentially influencing behavioral responses to environmental visual stimuli.

**Circadian rhythms in adults.** Our paradigm enabled us to track the circadian features of adult behaviors and their emergence in pups. Studies on gerbil circadian rhythms indicate that light conditions can affect behavior [35,60,61], with a crepuscular pattern for most behaviors except drinking [35]. We found adults were active during both light and dark cycles, with more behavioral activity during the day time and a substantial increase of activity prior to dark cycle onset. This

pattern is reminiscent of crepuscular behaviors in field studies, although not identical. Two significant differences with field studies is that our lab paradigm contained no risks, such as the presence of predators, and offered ad libitum food and water resources. We also observed diurnal behavior patterns only in the amount of distance traveled with other behaviors showing weak or no crepuscularity. We note that increased activity during the day is consistent with findings of greater maternal attention to pups during the day than at night [18,19].

**Development of a diurnal pattern in pups.** Beginning at approximately P26, pups began to display diurnal patterns in all behaviors with peaks at night cycle onset. However, in contrast to adults, these older pups also displayed substantial diurnal changes in activity as light-change switches approached across most behaviors (note: pup behavior was weak in the pup–pup approach behavior – similar to adults). This suggests that pups, unlike adults, had stronger drives to increase their activity before light onset than adults. Considering all periods of development, however, we found that some behaviors such as open exploration time and pup–pup time spent together changed much less between early and late development as compared to food hopper region and waterspout proximity. We sought to use the heterogeneity in food hopper proximity in particular – as it showed the largest differences over time – to agnostically detect different developmental stages for circadian rhythms, and identified P16–19, P20–27, and P28–29 as the most likely partitioning of development into three stages.

**Limitations of our work and directions for future studies.** Our enriched and enlarged home cage environment enabled the tracking of several types of behaviors. However, although the cage area was approximately 5–6 times the size of animal facility cages, it was significantly smaller than gerbils' natural habitat, and lacked an underground burrow. In particular, providing access to an underground compartment during development permits the maturation of normal escape behavior [62], suggesting that foraging area size alone is insufficient for the emergence of a full behavioral repertoire. Therefore, developmental trajectories may differ with orders of magnitude larger areas to explore. Additionally, we did not carry out highly specific behavior classifications such as grooming, fighting, playing, and sexual interactions, which would require significantly more complex machine vision tools than those developed in this study. We also substituted one day of our study due to cage cleaning with the average behaviors on neighboring days, and this may have affected some of our metrics which relied on stable or rapidly increasing behaviors over days. Our environment did not capture other behaviors observed in the wild such as gerbils' increased tendency for exploration towards novelty [63] and towards "open-arms" of mazes (compared to rats and mice; [64]). Gerbils also show increased unpredictability in their escape behaviors as compared to other rodents [65], and it would be interesting to observe this escape behavior dynamics by simulating predators within our environment. Beyond home cage environments, we propose that contemporary machine learning methods are improving substantially and it may be possible to carry out similar studies in "limited size" environments such as scientifically-purposed barns [66,67].

**Understanding development from the micro to the macroscale.** Our work builds on a series of technically and computationally advanced paradigms that quantify behavior over days to months. A small group of adult mice have been monitored continuously for 3–4 days in 50–70 cm² chambers using video recordings, permitting the acquisition of multiple behaviors that can be modeled to reveal group dynamics [68–70]. In contrast, groups of 10–40 mice have been monitored continuously for up to 3 months within 1–2 m² environments using radio frequency identification transponders, thereby revealing the emergence of individuality [71,72].

Similar to these studies, our approach supports the viability of studying rodent social behavior across orders of magnitudes of temporal resolution during highly dynamic periods of behavior and development in family groups of rodents. Our findings reveal novel time courses for development in rodents, including dynamic periods of behavior acquisition and the development of circadianity. We view that increasingly ethological (e.g., group and family housed animals) and naturalistic (e.g., larger home enclosures with increased enrichment opportunities) behavior paradigms in the laboratory environment is an adequate and complementary method for the characterization of behavior and CNS development in health and disease.

## Methods

### Experimental animals

Three gerbil families (*Meriones unguiculatus*, $n = 6$ per family: 2 adults, 4 pups) were used in this study (Charles River). All procedures related to the maintenance and use of animals were approved by the University Animal Welfare Committee at New York University, and all experiments were performed in accordance with the relevant guidelines and regulations (protocol 2020-1112).

### Audio recording

Four ultrasonic microphones (Avisoft CM16/CMPA48AAF-5V) were synchronously recorded using a National Instruments multifunction data acquisition device (PCI-6143) via BNC connection with a National Instruments terminal block (BNC-2110). The recording was controlled with custom python scripts using the NI-DAQmx library (https://github.com/ni/nidaqmx-python) which wrote samples to disk at a 125 kHz sampling rate. In total, 13.084 TB of raw audio data were acquired across the three families. For further analyses, the four-channel microphone signals were averaged to create a single-channel high-fidelity audio signal.

### Home cage enclosure

We used an enlarged home enclosure (W × L × H: 35.6 cm × 57.2 cm × 38.1 cm) with ad lib access to a food hopper and a water source.

### Video recording

We recorded video using an overhead camera, FLIR USB blackfly S (FLIR USA), at 25 frames per second. We recorded individual periods of approximately 20 min (cohort 1) or 60 min (cohorts 2 and 3) following which video and metadata were saved for offline use. The saving period resulted in approximately 10% downtime between recordings.

### Night:day cycle

Daytime began at 08:00 and night began at 20:00 during which an infrared light was activated.

### Animal shaving patterns

We used a distinct shaving pattern across all our cohorts (as described in Fig 1b). We opted for use of shaving patterns instead of fur bleaching, implanted trackers or body piercing as we viewed shaving as a less intrusive method.

### Tracking—Features

We manually labeled 1,000 frames using the SLEAP (Pereira and colleagues [30]) labeling workflow. The skeleton consisted of 6 nodes: nose, spine1, spine2, spine3, spine4, and spine5; and 5 edges: nose to spine1, spine1 to spine2, spine2 to spine3, spine3 to spine4, and spine4 to spine5. Additionally, we labeled each instance with the animal's identity class as female, male, pup1, pup2, pup3, and pup4 to enable tracking unique identities. Due to idiosyncratic differences in shaving patterns, lens distortion and light conditions, each cohort and light condition (i.e., day and night) was labeled and trained individually, resulting in 6 feature models. The final results were trained using SLEAP bottom-up ID models that achieved 97% accuracy of adults and 91% for pups.

### Track clean up—features

We developed an ID-switch error detection method to detect and correct some of the SLEAP ID-switch errors. We defined ID switch errors as those for which track segments were swapped between two animals. These errors were identified by

evaluating sequential segments of tracks produced by SLEAP and determining whether (i) a large jump in track location occurred; and (ii) whether another segment fit better (usually within <5 pixels of distance). We additionally implemented an interpolation method that connected SLEAP segments <3 cm apart that belonged to the same animal. A custom python pipeline was developed to carry out this correction.

### Tracking—Huddles

We manually labeled 1,000 frames using the SLEAP (Pereira and colleagues, [30]) labeling workflow. The skeleton consisted of 1 node: huddle; and no edges. We identified a huddle as a group of 2 or more animals grouped together in a stable location, which were usually places that animals huddled in for the entire recording period or several sequential days. Due to idiosyncratic differences in shaving patterns, lens distortion and light conditions, each cohort and light condition (i.e., day and night) was labeled and trained individually, resulting in 6 huddle models. We additionally implemented an interpolation method that connected SLEAP segments for huddles that were less than 30 s apart.

### Bedding changes and exclusion of time points

We provided an interpolated measure of behavior on days of bedding changes. Briefly, on cohorts 1 and 2 we averaged the behaviors between P23 and P25 instead of using P24. On cohort 3 on P18 we averaged the behaviors between P17 and P19 instead of using the P18 day as we observed a substantial difference in the behavior just for that day.

### Cross-validation of behavior tracking

We implemented 10-fold cross-validation to evaluate the precision of SLEAP identity tracking against a human annotator. Briefly, for all feature NNs we split our human labeled frames dataset (1,000 labeled frames) into 950 – training and tested on the hold out of 50 frames. We repeated this 10 times by randomizing the training set frames and calculated the average identity accuracy errors across adults and pups.

### Convex hull overlap metric

We defined overlap volume between behaviors as the intersection between the convex hull generated by individuals in each type of behavior relative to the other behaviors. We calculated the overlap using principal component analysis (PCA) as the intersectional volume of the first 3 principal components (using the python vedo package) or – when this was not possible due to insufficient points to form a polyhedron – the first 2 principal components into the PCA. The data input into PCA was [n_animals, n_time_points] = [12,15] where the values were normalized to adult levels. We viewed the overlap volume as better at capturing and representing the inter-behavior similarity by providing a metric that quantifies when some of the individual behaviors were not distinguishable between behavior types. We note that comparing the individual behaviors at each time point was not practical due to only 3 samples in the behavior of the adult female and male and male pup. We also note that we did not use more than 3PCs (or the full dimensionality of the distributions) to compute these similarity metrics as algorithms to compute multidimensional overlapping volumes are not available. Furthermore, the dimensionality in our study would likely yield vanishingly small overlaps.

### Rapid development and stabilization periods

We defined a rapid development period as days that had distributions that were significantly different than the previous 2 days. We indicated these periods with triangles in the plots in Figs 2 and 3. We thus pooled behaviors in groups of 2 days and compared them following groups. We used two days as a more conservative way to account for higher variability – than applying smoothing or other filtering techniques on the raw data. We used a 2 sample Kolmogorov-Smirnov test as a more conservative measure of significance. For clarity, at each day, we pooled data from individuals (e.g., 12 pups or 6

adults) and calculated the statistics across these values. We used the python package *statsmodels* method *sm.stats.mul-tipletests* for the Benjamini–Hochberg multiple hypothesis testing adjustment and reported *p* values < 0.05 as statistically significant.

### Heuristics-based behavior classification

We implemented heuristics to compute the location of animals near the food hopper or waterspout and each other. For the food hopper we used an ROI of size 19.50 × 10.50 cm. For waterspouts we used an ROI of size 8.50 × 8.00 cm. For inter-animal proximity we used a distance of <5 cm separation and a minimal proximity time >200 ms. Animal locations were calculated as the median centroid over ($x$, $y$) locations. For control and comparison to ROIs in food and waterspout regions we also selected ROIs in random locations of the cage. We found that ROIs that were randomly placed had little occupancy (limiting statistical analysis) and did not have the dynamics of ROIs in the food and waterspout regions.

### Computation of group socialization time

We calculated the amount of time pups and adults were socializing as the time they spent within 5 cm of each other. This method was used for both pups and adults, for example, to compute how much time each age group spent near groups of 2 or 3 other animals (i.e., <5 cm from nearby gerbils). The relative time computation was calculated similar to other steps, namely, by computing the socialization time for adults near other gerbils including other adults and pups and using this as a baseline.

### SimBA-based behavior classification

We defined an approach as an animal moving towards another – beginning when the initiator starts moving and ending when it stops moving (at most one gerbil length away from the other) or makes physical contact. We manually annotated 300 approaches (4,578 frames) in 22 videos and implemented a modified version of SimBA [31] for classifying pair-wise animal approach behaviors. We calculated the threshold for approach behavior as the 99% percentile of the prediction distribution. We verified that detected approaches matched with qualitative approaches.

Descriptive statistics of animal movements, angles, and distances in sliding time-windows were calculated using run-time optimized methods available through the SimBA API and used in a downstream random forest classifier. We randomly sampled annotated frames from separate events in training and test set to reduce time-series data leakage. Within the training set, we further balanced the data by under-sampling the number of non-event observations to a 1:1 ratio of the event observations. Classifiers were validated and optimal discrimination threshold determined by evaluating performances on novel held-out videos excluded from the training and test sets.

### Multi-animal social state networks

We developed an approach to quantify unique social configuration states and their dynamical transitions. We first discretized the cage space into 5 × 4 squares of approximately 11 cm × 9 cm each – yielding a 20 Dimensional state space. At each time point in our recordings we calculated the occupancy of each location as the number of animals at that location. This yielded an $N_{videoframes}$ × 20D matrix which in general had a few hundred or thousand unique 20D vectors per day. Each 20D vector thus contained integers from 0 (no animal in the location) up to 6 (i.e., all animals were at the grid location). Connectivity matrices for each day were then calculated by connecting each of the videoframe nodes to the one occurring subsequently. We used the connectivity matrices to generate 14 networks for each day of our study and for a variety of animal configurations. For clarity, we calculated 5 types of networks: single pups (i.e., 1 animal only), pairs of animals (the 2 adults and all pairwise combinations of pups), all-pups (i.e., all 4 pups) and all-animals (all 6 animals). (Fig 5a, 5b; see also "Methods"). This enabled us to generate a network-based analysis where each time point in our recording generated a 20 Dimensional (20D) social state vector and we could generate a network connectivity matrix by connecting all social

states that occurred sequentially (Fig 5c; see also "Methods"). The network connectivity matrix enabled us to generate graphs that represented the social state dynamics at each day in our cohorts (Fig 5d).

**Automatic detection of circadian rhythms development.**

We developed a computational method to evaluate whether pups had more discrete transitions from non-circadian to circadian rhythms-driven behaviors. The method was applied to food hopper region proximity data – and an assumption of three developmental periods based on qualitative observations in our study defined as: an initial period D1, a middle period D2 and a final period D3. Based on our qualitative observations we initialized these periods as: P16–P20, P21–P25 and P26–P29. We next sought to determine whether the boundaries on these periods, i.e., $t1$ and $t2$ corresponding to P20 and P25, respectively, could be quantitatively defined via an optimization function. The optimization function sought to maximize the difference between the individual animal circadian trajectories in each of these periods. Our algorithm identified $t1 = 19$ and $t2 = 27$ as the dates that create the maximum differences between pup presence in the food hopper region, i.e., D1: P16–P19, D2: P20–P27, and D3: P28–P29. (Fig 5h). We noted that different behaviors or priors, e.g., a single transition may provide slightly different developmental windows.

**Cohort and animal group sizes power analysis**

To achieve a power of at least 80% with a significance value of 0.05 we estimated the required sample sizes for our work are 3 individuals in each group of pups and adults – whereas we had 12 and 6, respectively. Briefly, we estimated these values based on existing literature that shows P16 gerbil pups having little to no movement (they are still in the nest for most of the time) and a small variance of 10% relative to the adult (which was our baseline) and the adult having approximate variance of 20%. Thus, for P16 and early development, 3 individuals in each group were sufficient, with 12 and 6 animals in each group providing power of approximately 90% or better for both 0.05 and 0.01 significance. As we trained our neural networks to identify pups and adults across all days of our study (P16 to P29) our cross-validation analysis applies to all age groups – not just early development – and we found strong evidence that the machine-vision tools generalize sufficiently to keep the same animal identity over the entire window of development in our study. In sum, our data and analysis support that groups of approximately 3 individuals each would provide sufficient power for this study.

**Computation of standard deviation**

Standard deviation reported in all results in brackets is for all pups, i.e., $n = 12$. The values for each pup was obtained by averaging the behavior of the 2 adults within each cohort.

**Supporting information**

**S1 Fig. 10-fold cross-validation evaluation of accuracy of animal tracking algorithms.** We labeled 1,000 frames per cohort under both day and night conditions (total 6,000 human labeled frames across three cohorts), and studied the accuracy of the animal tracking algorithm (SLEAP) as a function of the number of frames ($x$-axis) trained. We performed this analysis 10 times per training set size by shuffling the human labeled datasets (except the 1,000 – dataset where we held out only 50 frames and trained on 950). Accuracy of unique animal identification improves asymptotically with the number of human labeled frames in the dataset.
(TIF)

**S2 Fig. Real versus identity shuffled development trajectories.** To visualize the uniqueness of the developmental trajectories, we plotted water **(a)** and food **(b)** consumption behaviors when pup versus adult identities for correct or "real"

identity tracking versus identity shuffled condition. Shuffling identities results in behaviors appearing closer to adults and with higher variance (shading is SEM).
(TIF)

**S3 Fig. Shuffling identity decreases behavior trajectory uniqueness.** Similarity of intra- versus inter-behavior trajectories over time visualized with principal component analysis (PCA). The individual developmental trajectories (14 Days = 14 Dimensional) for all pups ($n$ = 12) is visualized using the first two PCs of these datasets. **(a)** When taking into account unique identities developmental trajectories (points) and convex hull (enclosed areas) for water and food proximity in PCA space show significant separation. **(b)** When the analysis shown in panel **(a)** is repeated, but with unique animal identity shuffled, substantial overlap is observed.
(TIF)

**S4 Fig. Correlation of real versus shuffled development trajectories.** Distributions for all pair-wise individual pup developmental trajectories over 14 days of development are shown for proximity to water and food sources. Within each behavior class, the correlations are much higher for real versus identity-shuffled data. Between behavior correlations, such as water versus food proximity, were lower than within behavior correlations, but significantly higher than those computed for shuffled conditions. We note that correlation between food and water development trajectories was relatively high because both behaviors are generated by the same causal factors, including the increased drive to seek independent sustenance, which unfold on a similar time scale.
(TIF)

**S5 Fig. Inter-behavior pairwise correlations.** Comparison of food and water proximity to other behaviors yields positive or negative correlations. This result is due to the Pearson correlation between a coarse measure of similarity that relies on the relative value of points in a time series to the mean. As many of the behaviors increased or decreased monotonically in similar ways, we did not view these correlations as a sufficiently sensitive method of analysis in our study.
(TIF)

**S6 Fig. Cohort similarities show partial within group idiosyncrasies.** We performed a reanalysis of the data plotted in Fig 2C, but with members of the same cohort displayed as different symbols. There is some grouping of gerbil pup behaviors by cohort number (we used a 2D PCA and a different projection angle to improve visualization). This partial clustering is indicated by the proximity of trajectories (i.e., individual points) for gerbils from similar cohorts (denoted by symbol). However, our primary conclusion is preserved: pups from the same or different cohorts still preserve distinct behavior trajectories (in this higher-dimensional space) when compared across behaviors – and the findings are not drive by single cohort idiosyncrasies.
(TIF)

## Acknowledgments

We thank Garet Lahvis, Mark Blumberg, Kenny Kay and Paola Cerrito for their helpful suggestions or comments on the manuscript.

## Author contributions

**Conceptualization:** Catalin Mitelut.

**Data curation:** Catalin Mitelut, Marielisa Diez Castro, Ralph Emilio Peterson, Madeline Gamer.

**Formal analysis:** Catalin Mitelut, Marielisa Diez Castro, Ralph Emilio Peterson, Maria Goncalves, Jennifer Li, Madeline Gamer, Simon RO Nillson, Talmo D Pereira.

**Funding acquisition:** Catalin Mitelut, Dan H Sanes.

**Investigation:** Catalin Mitelut, Marielisa Diez Castro, Ralph Emilio Peterson, Maria Goncalves, Jennifer Li, Madeline Gamer, Simon RO Nillson.

**Methodology:** Catalin Mitelut.

**Project administration:** Catalin Mitelut.

**Resources:** Catalin Mitelut, Dan H Sanes.

**Software:** Catalin Mitelut, Marielisa Diez Castro, Ralph Emilio Peterson, Simon RO Nillson, Talmo D Pereira.

**Supervision:** Catalin Mitelut, Dan H Sanes.

**Validation:** Catalin Mitelut.

**Visualization:** Catalin Mitelut, Maria Goncalves, Madeline Gamer.

**Writing – original draft:** Catalin Mitelut.

**Writing – review & editing:** Catalin Mitelut, Marielisa Diez Castro, Ralph Emilio Peterson, Dan H Sanes.

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
