## [Editor Report · Decision Letter 0]

18 Jan 2025

Dear Catalin,

Thank you for submitting your revised manuscript entitled "Developing gerbils exhibit structured social behaviors prior to the emergence of autonomy" for consideration as a Research Article by PLOS Biology.

Your revisions has now been evaluated by the PLOS Biology editorial staff. Sorry for the delay; the Academic Editor has been travelling, so we don't have advice from them, but I'm writing to let you know that we would like to send your submission out for re-review.

However, before we can send your manuscript back to reviewers, we need you to complete your submission by providing the metadata that is required for full assessment. To this end, please login to Editorial Manager where you will find the paper in the 'Submissions Needing Revisions' folder on your homepage. Please click 'Revise Submission' from the Action Links and complete all additional questions in the submission questionnaire.

Once your full submission is complete, your paper will undergo a series of checks in preparation for peer review. After your manuscript has passed the checks it will be sent out for review. To provide the metadata for your submission, please Login to Editorial Manager (https://www.editorialmanager.com/pbiology) within two working days, i.e. by Jan 21 2025 11:59PM.

Kind regards,

Roli

Roland Roberts, PhD

Senior Editor

PLOS Biology

rroberts@plos.org

---

## [Decision Letter · Decision Letter 1]

11 Mar 2025

Dear Catalin,

Thank you for your patience while we considered your revised manuscript "Developing gerbils exhibit structured social behaviors prior to the emergence of autonomy" for publication as a Research Article at PLOS Biology. Your revised study has been evaluated by the PLOS Biology editors and the original reviewers. As mentioned in my earlier email, we were unable to obtain input from the Academic Editor in a timely fashion, so we're sending the decision anyway, and will pass on any further requests that they have in due course.

You'll see that reviewer #1 reiterates their previous point that ideally the sample size should’ve been larger, but won’t hold up publication. S/he wants you to incorporate the rebuttal Figs into the paper (or its supplement), cite a number of comparable studies, define “solitary behavior,” remove mention of ultrasound recordings, clarify a number of statistical methods and other points, and improve presentation. Reviewer #2 also makes the point that while it’s good to have removed the underpowered sex difference analysis, increasing sample size would’ve been better. S/he wants more clarity around your stats and interpretation, and has suggestions for developing the Discussion, plus a couple of possible analyses. Reviewer #3 is largely satisfied and only has one minor request.

In light of the reviews, which you will find at the end of this email, we would like to invite you to revise the work to thoroughly address the reviewers' reports.

Given the extent of revision needed, we cannot make a decision about publication until we have seen the revised manuscript and your response to the reviewers' comments. Your revised manuscript is likely to be sent for further evaluation by all or a subset of the reviewers.

**IMPORTANT - SUBMITTING YOUR REVISION**

*Re-submission Checklist*

*Published Peer Review*

*PLOS Data Policy*

*Blot and Gel Data Policy*

Sincerely,

Roli

Roland Roberts, PhD

Senior Editor

PLOS Biology

rroberts@plos.org

REVIEWERS' COMMENTS:

Reviewer #1:

The revised manuscript provides a thorough analysis of gerbil behavioral development using machine vision and continuous tracking in home-cage environments. It offers valuable insights into the emergence of social and autonomous behaviors. The authors have effectively addressed many of the concerns raised in the prior review. Rather than increasing the sample size, they have opted to remove the sex-based analyses due to limitations in experimental design and sample size. That being said, I continue to believe that for a purely descriptive study—one that does not test a specific hypothesis through a priori experimental design—a larger n is necessary to draw strong conclusions. In this case, the sample size could have been larger. However, I will not oppose publication on this basis. Several points still require clarification and revision (see below). Once these issues are addressed and the text is revised accordingly, I believe the study will be suitable for publication.

1) Incorporation of Supplementary Figures: The rebuttal letter references supplementary figures that provide interesting insights. However, unless I missed something, I do not see references to supplementary material in the main manuscript. Why is this material not included in the paper, or at least some of it? If these figures strengthen the main findings, it would be beneficial to integrate them.

2) Long-Term Tracking and semi natural environment References: The manuscript would benefit from referencing additional studies that have tracked behaviors over extended periods in semi natural environments, even if it is not an analysis at a sub second scale. Relevant works include among other Shemesh et al. (2013), Freund et al. (2014), Torquet et al. (2018), De Chaumont et al. (2019), Forkosh (2019)…. Citing these studies does not diminish the novelty of the authors' recordings but rather strengthens the context of long-term behavioral tracking in rodents.

2) Definition of Solitary Behavior in Introduction: The term "solitary behavior" should be clearly defined, particularly since "agency" is discussed earlier. Here is my understanding: Solitary behavior refers to actions that an individual performs alone, without direct social interaction. But it is not non-social behaviour. For example, an individual may develop self-sufficiency or problem-solving skills independently while still being part of a social group.

3) Ultrasonic Recording Mention in Methods: The sentence "Continuous video recordings were obtained from an overhead camera (24 fps), and audio recordings were obtained from four ultrasonic microphones (125 kHz sampling rate)") is redundant and should be removed as it a repetition of what has been said before. Furthermore, since ultrasonic recordings are not analyzed in the paper, all references to them should be removed from methods section, and image (Figure 1) ?.

4) If percentages are used in the text (for example page 7: We found that pup distance traveled per day started at 8.72 ± 1.71 % of adult levels (or 8.01 ±1.53 meters/day) and increased continuously until P26 where it stabilized near adult levels at 85.90 ± 13.08 % then figures should also use percentages rather than values relative to adult averages—or vice versa.

6) Clarification of Sample Size (n=12): When variances or standard deviations are estimated on 12 individuals, explicitly state n=12 in figures. Also, clarify whether the reference of 1 is obtained by averaging the two adults per cage.

7) Pairwise Social Exploration and Nest Time: Was pairwise social exploration analyzed in relation to time spent in the nest? Were day and night phases accounted for? Additionally, were sleep periods near each other counted as social interactions? These points need clarification.

8) Calculation of Time Spent in Groups: Provide details on how time spent in groups of adults was calculated. In the sentence: "We found that pups socialized in groups of three gerbils 6 to 7 times more than adults did in early development when they left the nest (i.e., as early as P16) and leveled off around 2 times as much by P29 (Fig 3h)." What is the reference for adult grouping? Since there were only two adults per cage, it could not have been groups of three adults—so what was the comparison based on? Clarify the baseline used for this analysis.

9) There is no reference to Figure 3j in the text.

10) The 3D PCA space used in Figure 2c is not sufficiently defined in the Methods section. If my understanding is correct, the analysis starts with an m × n matrix, where m (rows) corresponds to all pups (N=12) across four behaviors (4 × 12 rows), and n (columns) represents 14 days. The data used appears to be from Figure 2b (normalized to adult levels). If so, confirm in the Methods section. Also, does the repeated use of the same animal for four rows in the matrix introduce unexplicit dependencies in the PCA analysis?

11) Clarification of "Distinct Trajectories" Statement: The sentence "This supports that the development of individual behaviors have distinct trajectories - rather than being explained by coming from pups in the same cohort". Thanks for adding this sentence but to fully support this claim, an analysis of trends within each cage would be needed. If all pups in a given cage show similar trends, then cage environment might be influencing behavior. A simple way to show that would be to color members of the same cohort differently in figure 2c to confirm that points are not clustered by family?

12) The conclusion states: "Our findings reveal novel time courses for development in rodents, including dynamic periods of behavior acquisition, sex-based behavior differences, and the development of circadianity." Removed "sex-based behavior differences".

Reviewer #2:

Positive comments:

Removing underpowered study of sex differences is an improvement, but adding in more gerbils would have been preferable.

The new discussion sections on puberty and maturation of spatial-cognition, motor, threat and motivation systems adds richer, more balanced context.

Concerns:

Overall, the statistics in results and figures could be more explicitly explained and reported. Currently, it is hard to fully follow how conclusions were supported and what comparisons were made and deemed to be significant.

"We note that both distance traveled and environment exploration time underwent rapid development for most of the days between P17 and P25." Did the authors want to back this up by reporting the KS tests used?

In the figures there are delta(triangle) symbols, but they are not explicitly linked to a significant KS test by this comment "with indication of rapid development days" in Fig 2. The symbols are in Fig. 3 legend too but are not defined there.

"Similarly, pups were followed out of the nest by parents significantly more than parents followed other parents on P16 at 151 ± 0.17 % (and overall followed 76.09 ± 8.57 % of the time), and this nest following behavior stabilized also around P24 at 107 ± 12 % (or 53.53 ± 5.88 %) (Fig 3b)" Statistical tests done here unclear.

Authors in results: "This suggests that increased group sizes could be supportive of environmental exploration and that single animals or smaller groups tend to use more stereotyped or repeated trajectory patterns." Does this correct for chance?

Circadian section is also lacking statistics for group comparisons. Are there any statistics? There seemed to be differences in peaks in night for one period for adolescents. Were comparisons made so age period was paired between adult and adolescent groups (since activity one might affect the other)? Unclear to be How is circadian behavior (or non circadian) behavior defined and tested?

Interpretation of data varies by section. Authors in Results note 100% more time in food hopper region in late P20s pups. They interpret this in results with "This suggests that in addition to learning the nutritional value of the food hopper region, pups may also value this space as a socialization space significantly more than adults (see more below)." (Alternatively, do they just need to eat more? Was this due to 2nd period nighttime eating seen in circadian figures?)

Later in the discussion they say "These findings provide independent confirmation that the pups' increase of food hopper and waterspout proximity were driven, in part, by consumption behaviors and a decreased dependence on the adult female."

Discussion comments

In the discussion, there is almost no development of the social ideas emphasized earlier in the manuscript and in the abstract (claim "Our work supports a model in which early and sustained social interactions may be supportive of solitary exploration for physiological needs"). If the authors want to invest more in this social support idea, I think they might add in studies that posit social influences on behavior in other species? I think this is in literature coming from primates (great apes), dolphins, maybe elephants who also show youth groupings weaning to juvenile phase. If gerbils have complex social structure this could be interesting link.

https://doi.org/10.1016/j.anbehav.2017.04.011

Claim in discussion: "This supports that the development of individual behaviors have distinct trajectories - rather than being explained by coming from pups in the same cohort."

This is an interesting point that could perhaps be more thoroughly supported in explicit terms. For example, we do not get to see any plots of the three different cohorts next to each other and the order of their relative developmental sequences. Also were pup weights (as a proxy for developmental advancement) comparable between cohorts at the start of the experiment? Does this correspond to temporal rank in terms of the unfolding of events?

Relevant new study that might be cited Power et al., 2025 Animal Behavior https://doi.org/10.1016/j.anbehav.2025.123084

Minor:

Typo: Worman=workman 2013

Between-sex statistical differences tests still in methods

Reviewer #3:

[identifies herself as Ewelina Knapska]

I am fully satisfied with the authors' amendments. This is a solid study that offers valuable insights into the early development of gerbils and presents a useful methodological approach. I congratulate the authors on their work.

I have just one minor comment: in the Discussion section (p. 20), there is a reference to sex-based behavioral differences, but since this part of the results has been removed, the statement no longer fits.

---

## [Editor Report · Decision Letter 2]

7 Jul 2025

Dear Catalin,

Thank you for your patience while we considered your revised manuscript "Developing gerbils exhibit structured social behaviors prior to the emergence of autonomy" for publication as a Research Article at PLOS Biology. This revised version of your manuscript has been evaluated by the PLOS Biology editors and the Academic Editor.

Based on our Academic Editor's assessment of your revision, we are likely to accept this manuscript for publication, provided you satisfactorily address the remaining points raised by the Academic Editor, and the following data and other policy-related requests.

IMPORTANT - please attend to the following:

a) Please modify the Title slightly (in both the manuscript file and in the EM metadata) to include the methodological approach. We suggest: "Continuous monitoring and machine vision reveals that developing gerbils exhibit structured social behaviors prior to the emergence of autonomy"

b) The Academic Editor has kindly gone through your manuscript, and has made a large number of detailed recommendations (see the foot of this email). S/he said "I do think there’s a fair bit of work needed to tidy this into shape from a paper-craft perspective (lots of methods and results and discussion and interpretation all mixed up, I don’t mind a little contextualisation in longer papers but this does come across as a bit of an inconsistent hodgepodge at the moment, and there is direct repetition with sections that remain in the Methods). There are a few sections with a clearer polish in the discussion - but there is a fair bit of work to be done to showcase their interesting findings in a light that the reader can easily follow along with." and "at times there was so much ambiguity in what they were saying that it was really challenging for readability." We strongly recommend that you address all of the Academic Editor's requests and get a colleague who is a native English speaker to give the manuscript a final go-through.

b) Please supply the protocol approval number from the NYU Animal Welfare Committee.

c) Please supply a link to your funding agency in the Financial Disclosure Statement.

d) Please address my Data Policy requests below; specifically, we need you to supply the numerical values underlying Figs 1D, 2BCD, 3BDFHI, 4FHJ, 5BCDEFGH, S1, S2AB, S3AB S4, S5, S6, either as a supplementary data file or as a permanent DOI’d deposition (e.g. as part of your Dryad depositions).

e) I note that although you already have two associated Dryad depositions, these are currently not accessible. Please provide a reviewer link so that we can check your compliance with our data policy.

f) Please cite the location of the data clearly in all relevant main and supplementary Figure legends, e.g. “The data underlying this Figure can be found in S1 Data” or “The data underlying this Figure can be found in https://datadryad.org/XXXXX"

g) Please make any custom code available, either as a supplementary file or as part of your data deposition. I note that you mention "custom postprocessing code available at https://github.com/catubc/gerbil" - because Github depositions can be readily changed or deleted, please make a permanent DOI’d copy (e.g. in Zenodo) and provide this URL.

We expect to receive your revised manuscript within two weeks.

*Published Peer Review History*

*Press*

Sincerely,

Roli

Roland Roberts, PhD

Senior Editor

rroberts@plos.org

PLOS Biology

ETHICS STATEMENT:

-- Please include the full name of the IACUC/ethics committee that reviewed and approved the animal care and use protocol/permit/project license. Please also include an approval number.

-- Please include the specific national or international regulations/guidelines to which your animal care and use protocol adhered. Please note that institutional or accreditation organization guidelines (such as AAALAC) do not meet this requirement.

DATA POLICY:

Regardless of the method selected, please ensure that you provide the individual numerical values that underlie the summary data displayed in the following figure panels as they are essential for readers to assess your analysis and to reproduce it: Figs 1D, 2BCD, 3BDFHI, 4FHJ, 5BCDEFGH, S1, S2AB, S3AB S4, S5, S6. NOTE: the numerical data provided should include all replicates AND the way in which the plotted mean and errors were derived (it should not present only the mean/average values).

CODE POLICY

SPECIES INDICATED IN THE ABSTRACT?

- Please note that per journal policy, the model system/species studied should be clearly stated in the abstract of your manuscript.

DATA NOT SHOWN?

COMMENTS FROM THE ACADEMIC EDITOR:

I’ve done a clarity/typo level set of suggested edits below - I do think that the work would benefit substantially from a proper polish beyond these, but hopefully that’s a helpful start for them:

A few queries/non-linguistic points:

Line 107 -> 0.91 true positive rate for pups, suggests a 10% error rate in identification in their frames, given how few individuals there are (and how many frames) this is a little concerning. I’m not sure if they assessed whether this was randomly distributed, or if there were particular errors (e.g., two individuals were swapped systematically more frequently, which would be more problematic).

Typesetting check -> they’ve sometimes added bolding, I’m not sure if this is for emphasis, but I’d suggest that these are all removed (except titles etc.).

Line 160: there is an inference here about the social importance of the space, relative to adult use (as well as the nutritional value) but I’m not sure on what basis that is drawn? Is it only that they’re there twice as often as adults? Couldn’t that be to do with eating patterns (feeding more slowly? Etc.)

In almost all cases they give qualitative descriptions of patterns in the data, but I’m not sure why they don’t test some of these in a quantitative manner? e.g., all the values on page 10 of pup-pup vs pup-adult interaction time could be tested with very straight forward measures that would let them say with greater confidence whether there was anything there beyond potential patterns.. I hate to ask for additional tests at this point, but given regular use of terms like ‘significantly more’ they seem interested in making these claims. If they’d rather not, then I think more work is needed to de-emphasise the behavioural findings and interpretation.

Line 377: you used social and solitary in the intro, do you mean the same thing here with social and autonomous?Line 406: I would avoid ‘advanced’ mammals, there aren’t really more/less advanced species

Line 407: Harlow conducted a long and infamous series of experiments on early social deprivation in primates

Line 461-465: a possible contribution to the variation with field studies would be the lack of predators in captive contexts..

Line 481: could the absence of underground areas also impact circadian rhythm development?

Line 591: dimensionality is indeed a curse, but perhaps there is a more formal way to refer to this..

Basic writing edits:

Line 41: social behavior -> social behavioral

Line 42: than anticipated -> than typically considered

Line 45: numerous survival and thriving goals -> goals allowing individuals to survive and thrive

Line 49: largely rare -> rare

Line 50: these two sentences don’t seem to follow from each other, clarification needed?

Line 54: family social bonds -> familial social bonds

Line 57: motor (REF) and -> motor (REF), and

Line 58: emerge -> emergence; an attraction -> the attraction

Line 63: what is ‘strong’ social structure’? -> perhaps strong social relationships

Line 64: We operationalise.. -> Here, we operationalise

Line 65: cooperation with other family members -> cooperation with other individuals

Line 66: with a pair-bonded -> with pair-bonded

Line 70/71: (P) 16, and while -> (P) 16 and, while

Line 76: leave -> left

Line 78: behavior classifiers (REF) and -> behavior classifiers (REF), and

Line 80: preceding -> precede

Line 81: if you say more, more than what? (More …….. than is typically recognised)?

Line 109: which are -> that were

Line 123: This -> These tracking results

Line 130: independently to -> independently of

Line 130: is it only the parents that are adults? If so cut adult, if not cut parent.

Line 138: relative to the adult -> relative to the adult mean.

Line 140: when it was -> in which it was; but also ‘it’ is ambiguous here, what does ‘it’ refer to?

General note -> most of your uses of ‘computed’ can/should be edited to ‘calculated’

Line 145: this sentence/method is unclear, I’m not sure what ‘this’ refers to, perhaps ‘of the variance’? Some clarification needed.

146: was more interpretable -> was more easily interpretable

147: distances which -> distances, which

Line 149: P26 where -> P26, where

Line 153: These open environment exploration findings -> these findings on open environment exploration

Line 157: to P21 at 0.45 ….. -> to 0.45 …… at P21

Line 158: between P22 until P26 stabilizing from P27 - P 29 -> between P22 and P26, and stabilizing from P27 to P29

Line 159: This -> This pattern (general note, unspecified this/these/it can be challenging for the reader to interpret)

Line 161: do you mean that they spent more time eating while at the hopper? (This phrase makes it sound like they spend more time now, because earlier in development they consumed more calories -> clarification needed).

Line 164: with pups spending -> in which pups spent

Line 164: adult averages -> adult average

Line 166: access -> exploration (ideal to keep consistency in terms, unless you separately measured access here?)

Line 167: spend -> spent

Line 168: relative to the adults -> of mean adult time

Line 175: This what?

Line 179: should decrease be decreased or decreases? (Unclear)

Line 181: but does not affect -> but this behaviour clustering does not appear to affect

Line 183: measures we -> measures, we

Line 183: distinct developmental -> distinct gerbil pup developmental

Line 185: cut adult

Line 186: out for this -> out to investigate the effects of this

Line 186: Thus, food and water -> Food and water

Line 201: 0.02 (or .. -> 0.02 of the adult mean (or

Line 203: more than -> for more time than

Line 203: 0.02 -> 0.02 of the adult mean.

Line 203: stabilized also -> also stabilized

Line 208: please remove the word significantly (you don’t seem to have tested this?)

Line 210 -> in several places in these results please specify where the value is ‘of the adult mean’ or ‘of adult-pup mean’ etc.Line 214: please remove the word significantly (you don’t seem to have tested this?)

Line 216: over all days -> across the full observation period

Line 217: show that while -> suggest that, while

Line 218: please remove the word significantly (you don’t seem to have tested this?)

Line 220: approaches with -> approaches, with

Line 221: this is closer to ~2 fold than to ~3 fold.. (the hours values are closer to ~3 fold)

Line 223-229: again please add ‘of adult means/average values’ etc. throughout for clarity.

Line 230 -> again as before, most uses of computed can be changed to calculated

Line 231: 6 to 7 fold more -> 6 to 7 times more frequently Line 232: off around 2 -> off at around 2

Line 234: This confirms -> This pattern suggests

Line 235: rather than being -> rather than partner preference being

Line 236: most behaviours, we find expected high -> behaviors, as expected, we find high

NB - please check for US/UK spelling consistency (e.g., behaviour vs behavior)

Line 237: shape of the behavior -> angle of the behavioral trajectories

Line 241, 242: e.g. -> e.g.,

243: substantially only at -> only increasing substantially at

Line 244: I’m not sure what integrity is used to mean here?

Line 262: This enabled -> This procedure enabled

Line 266: the social -> social

Line 269: suggest changing all use of animals in brackets to n=2, n=4, n=6

Line 270: sizes which represent -> sizes that represent

Line 272: please remove significantly (substantially?)

In general, please check over and remove all uses of the term significantly where there is no accompanying formal testing of the data.

Line 275: This suggests -> These differences suggest

Line 278: off around -> off at around

Please check all uses of computed -> computed implies a sophisticated calculation beyond the use of normal statistical methods (for which the term calculated can be used).Line 288: groups to be -> groups appears to be

Line 289: development or slightly decreasing suggesting -> development, or slightly decreasing, suggesting

Line 290: but rather were -> but were rather

Line 291: configuration -> configurations

Line 295: groups but not -> groups, but not

Line 295: We lastly computed -> Lastly, we calculated

Line 297: probability of a node to have -> likelihood that a node has

Line 302: configuration -> configurations

Line 305: 1.29 - suggesting -> 1.29; suggesting

Line 307: for exploration -> towards exploration

Line 312: This suggests -> These findings suggestLine 315: ‘may further extend this result into novel directions’ -> I think this is better phrased as ‘would provide a more robust test of this result.’

Line 321: values were -> values are

Line 326: This suggests -> These differences suggest

Line 329: night cycle and similarly with -> cycle, and similarly for

Line 339: I’m not sure what ramping refers to in this section here, and below (perhaps increases? Some uses specify ramping up, please clarify).

Line 352: Fig 5f) and -> Fig 5f), and

Line 353: et al 1982). -> et al 1982.)

Line 354: circadian behaviors? perhaps circadian patterns (Also unsure why circadian is italicised in multiple cases).

Line 358: being mostly absent from -> being largely absent in

Line 360: double space typo

Line 369: was the most striking developmentally -> provided the most obvious differences in development

Line 371: (e.g. assumes 3 -> (e.g., it assumes three

Line 385: are more similar when compared with -> were more similar to each other, than when compared with

Line 386: This suggests that many classes -> This pattern suggests that these classes

Line 387: perhaps -> random and/or individual processes? (Given that you didn’t find strong inter-individual differences)

Line 389: P26, when pups -> P26, at which time pups

Line 420: helpful to specify, ‘all social and foraging behaviours commence at near zero’ (it’s a little odd to say that all behaviors start at near zero)

Line 434: -> For example, as compared to adults, gerbil auditory

Line 452: It’s -> It is

Line 456: circadian rhythm effects -> effects of circadian rhythm

Line 461: perhaps ramping up? (Again, ramping on its own is unclear out of context).

Note: please check all uses of ‘ramping’ in results and discussion, in general there is a need to specify when this term is used (ramping up, ramping down, on ramping etc. etc.).

Line 474: compared to -> as compared to

Line 481: than the gerbil -> than gerbils’

Line 484: playing, sexual interactions which -> playing, and sexual interactions, which

Line 490: behaviors than other -> behaviors as compared to other

Line 505: orders of magnitudes greater?

Line 510: avoid undefined acronym in final sentence

Line 583: or when -> or - when

Line 584: into the PCA

Line 678: do you mean the comparison values for each pup or the values for adult averages? This sentence isn’t clear.

---

## [Editor Report · Decision Letter 3]

4 Aug 2025

Dear Catalin,

Thank you for the submission of your revised Research Article "Continuous monitoring and machine vision reveals that developing gerbils exhibit structured social behaviors prior to the emergence of autonomy" for publication in PLOS Biology. On behalf of my colleagues and the Academic Editor, Catherine Hobaiter, I'm pleased to say that we can in principle accept your manuscript for publication, provided you address any remaining formatting and reporting issues. These will be detailed in an email you should receive within 2-3 business days from our colleagues in the journal operations team; no action is required from you until then. Please note that we will not be able to formally accept your manuscript and schedule it for publication until you have completed any requested changes.

IMPORTANT: I've asked my colleagues to include the following very minor requests alongside their own:

1. Many thanks for providing your protocol approval number (2020-1112); please include this in the first paragraph of your Methods section.

2. Thank you for providing the links to your funding bodies; please include these in your Financial Disclosure statement.

Sincerely, 

Roli

Senior Editor

PLOS Biology

rroberts@plos.org